# Rock Phosphate-Enriched Compost in Combination with Rhizobacteria; A Cost-Effective Source for Better Soil Health and Wheat (*Triticum aestivum*) Productivity

**Motsim Billah [1], Matiullah Khan [2], Asghari Bano [3,\*], Sobia Nisa [4], Ahmad Hussain [5], Khadim Muhammad Dawar [6], Asia Munir [7] and Naeem Khan [8,\*]**

1   Department of Soil Science, University of Haripur, Haripur 21120, Pakistan; motsimbillah@gmail.com
2   Pakistan Agriculture Research Council, NARC, Islamabad 44000, Pakistan; mukhan65pk@yahoo.co.uk
3   Department of Biosciences, University of Wah, Wah Cantt. 47000, Pakistan
4   Department of Microbiology, University of Haripur, Haripur 21120, Pakistan; sobia@uoh.edu.pk
5   Department of Forestry, University of Haripur, Haripur 21120, Pakistan; ahmad.hussain@uoh.edu.pk
6   Department of Soil and Environmental Science, The University of Agriculture, Peshawar 25000, Pakistan; khadimdawar@yahoo.com
7   Soil and Water Testing Laboratory, Rawalpindi 46000, Pakistan; asiamunir800@gmail.com
8   Department of Agronomy, Institute of Food and Agricultural Sciences, University of Florida, Gainesville, FL 32611, USA
\*   Correspondence: bano.asghari@gmail.com (A.B.); naeemkhan@ufl.edu (N.K.)

**Abstract:** Organic materials from various sources have been commonly adopted as soil amendments to improve crop productivity. Phosphorus deficiency and fixation in alkaline calcareous soils drives a reduction in crop production. A two-year field experiment was conducted to evaluate the impact of rock phosphate enriched composts and chemical fertilizers both individually and in combination with plant growth promoting rhizobacteria (PGPR) on wheat productivity and soil chemical and biological and biochemical properties. The present study demonstrates significant increments in crop agronomic and physiological parameters with *Pseudomonas* sp. inoculated $RPEC_1$ (rock phosphate + poultry litter + *Pseudomonas* sp.) over the un-inoculated untreated control. However, among all other treatments i.e., $RPEC_2$ (rock phosphate + poultry litter solubilized with *Proteus* sp.), RPC (rock phosphate + poultry litter), HDP (half dose inorganic P from Single Super Phosphate-SSP 18% $P_2O_5$) and SPLC (poultry litter only); $RPEC_1$ remained the best by showing increases in soil chemical properties (available phosphorus, nitrate nitrogen, extractable potassium), biochemical properties (alkaline phosphatase activity) and biological properties (microbial biomass carbon and microbial biomass phosphorus). Economic analysis in terms of Value Cost Ratio (VCR) showed that the seed inoculation with *Pseudomonas* sp. in combination with $RPEC_1$ gave maximum VCR (3.23:1) followed by $RPEC_2$ (2.61:1), FDP (2.37:1), HDP (2.05:1) and SPLC (2.03:1). It is concluded that inoculated rock phosphate (RP) enriched compost ($RPEC_1$) can be a substitute to costly chemical fertilizers and seed inoculation with *Pseudomonas* sp. may further increase the efficiency of composts.

**Keywords:** available phosphorus; enriched compost; PGPR; poultry litter; rock phosphate; wheat

## 1. Introduction

The function of fertilizers for maximum crop production in under-developed countries is customary and well recognized. Nevertheless, the increasing prices of inorganic phosphate fertilizers and the extensive use of chemical fertilizers in agriculture, is also under debate due to environmental

concerns and for consumer health reasons [1]. Reduction of agrochemicals for crop production is of great concern for sustainable agriculture [2]. Moreover, inorganic phosphate fertilizers are not totally soluble in soil matrix due to precipitation reactions with ions of Al and Fe in acidic, and Ca in alkaline calcareous soils [3]. Moreover, high dose application of chemical fertilizers creates negative impacts such as changes in soil pH through alkalization and acidification, pollution of water resources through runoff, suppression of microorganisms and friendly insects, fixation of nutrients, degradation of soil structure due to increased decomposition of organic matter [4]. The research workers are required to look for substitutes to inorganic fertilizers [5], which are cost-effective and environmentally friendly. The use of rock phosphate (RP) as an alternative for P fertilizer is gaining attention in sustainable agriculture through microbial solubilization [6] and preparation of RP-enriched compost [7]. The mixing of RP with organic materials such as animal feces, plant residues and inoculation with acid-producing microbes may enhance P solubility from RP because when organic materials decompose, more soluble P is released due to the action of organic acids produced by the microbes [8]. The incorporation of organic residues either singly or in conjunction with a cheap source of mining element as rock phosphate may help to improve soil quality and productivity [9]. Rock phosphate enriched compost which was solubilized by phosphate solubilizing fungi and applied on a mung-bean crop, significantly enhanced yield and P-uptake [10].

Various RP enriched composts and inorganic fertilizers such as diammonium phosphate (DAP) were applied on wheat in a pot experiment. The data revealed that RP enriched composts showed no significant performance in the earlier stages of wheat growth but at maturity, it gave higher grain yield, nutrient uptake and increased fertility status of P and K in the soils [11]. Isolated phosphate-solubilizing fungi from phosphate mines of China were reported to have efficient biofertilizers and P solubilizers with the capacity to enhance the growth of wheat [12]. Colonization of soil by nonindigenous phosphate-solubilizing microorganisms depend both on their interactions with indigenous microorganisms associated with plants and their ability to utilize diverse substrates in soil [13]. The role of phosphate-solubilizing microorganisms in phosphate solubilization has been attributed mainly to their abilities to reduce the pH of the surroundings by the production of organic acids [13]. Preparing the RP-enriched compost with phosphate solubilizing microbes may not only compensate for the higher cost of manufacturing fertilizers, but also provide a sustainable source of available phosphorus to growing plants in alkaline soils [14].

Plant growth promoting rhizobacteria (PGPR) are important inoculants for integrated nutrient management [15] which help in dissolving inorganic P by excreting organic acids and chelation of P cations to release P in soil solution [16]. It was reported that there are several PGPR inoculants currently commercialized that promote growth either by suppression of plant disease, improved nutrient acquisition, or phytohormone production [17]. Generally, phytohormone in plants plays a vital role in cell division, proliferation, and differentiation, vascular tissue alteration, responses to light and gravity, general root and shoot architecture, seed and tuber germination, ethylene synthesis, vegetative growth processes, fruit development [18–20], initiation of lateral and floral organ and organogenesis [21], initiation of rooting, foliation and flowering [22], formation of lateral and adventitious roots [23], and increasing the growth of cambium and size of xylem cells [24]. Bacterial phytohormone production is widely distributed among plant-associated bacteria and is still considered the primary mechanism that enhances the growth and yield of plants [25].

PGPRs influence direct growth promotion of plants by fixing atmospheric nitrogen, solubilizing insoluble phosphates, secreting hormones such as IAA, GAs, and Kinetins besides ACC (1-aminocycloprapane-1-carboxylic acid) deaminase production [26], that helps in the regulation of ethylene. Amongst the majority of influential P solubilizers, bacterial strains from the genera *Pseudomonas*, *Bacillus*, *Rhizobium* and *Enterobacter* are of great importance. Application of phosphate solubilizing microbes in the production of compost can help to increase the interest of farmers to use organic phosphatic fertilizers in alkaline soils [14]. Therefore, this study aimed to evaluate the availability of phosphorus from RP enriched compost with the application of PGPRs and its

comparative effectiveness with inorganic fertilizers (Single Super Phosphate) on soil nutrient status, wheat growth and production.

## 2. Material and Methods

### 2.1. Experimental Site and Treatments

Two-year field experiments at National Agricultural Research Centre, Islamabad (73°70′ E and 33°39′ N with an altitude 610 masl during growing months Nov, 2010-May, 2011 and Nov, 2011-May, 2012), were conducted on wheat (var. GA-2002). Soil textural class of the experimental site was silty loam. The meteorological data during the growing season (2010–2012) of wheat is given in Table 1. Different composts being prepared during the previous experiments [27] were used in the study for their effectiveness to get better crop production. The treatments included; Control (Untreated un-inoculated); SPLC (Simple poultry litter compost); RP (rock phosphate 18.5% $P_2O_5$); $RPEC_1$ (rock phosphate + poultry litter solubilized with *Pseudomonas* sp. during composting process); $RPEC_2$ (rock phosphate + poultry litter solubilized with *Proteus* sp. during the composting process); FDP (Full dose inorganic P from Single Super Phosphate-SSP18% $P_2O_5$); HDP (Half dose inorganic P from Single Super Phosphate-SSP 18% $P_2O_5$). Treatments were applied at a rate of 100 kg P ha$^{-1}$, respectively from composts as well as from inorganic fertilizers on a total P basis during seed bed preparation. The nutrient status of different composts, is given in Table 2. The recommended dose of nitrogen at the rate of 100 kg ha$^{-1}$ was equally applied to each plot (4 m × 3 m) either from inorganic fertilizer (Urea-46% N) or compost on a nutrient basis. However, SPLC was applied at the rate of 4.5t ha$^{-1}$. There were three replications for each treatment. All the fertilizer treatments were applied to respective plots at the same time of sowing.

**Table 1.** Meteorological data during the growing seasons of wheat crop (2010–2012).

| Months | 2010–2011 | | | 2011–2012 | | |
|---|---|---|---|---|---|---|
| | Av. Temp (°C) | Rainfall (mm) | R.H (%) | Av. Temp. (°C) | Rainfall (mm) | R.H (%) |
| Nov. | 18.03 | 4.13 | 64.2 | 16.32 | 7.09 | 65.52 |
| Dec. | 18.18 | 26.97 | 64.89 | 19.23 | 0 | 62.74 |
| Jan. | 15.5 | 8.32 | 71.29 | 15.55 | 59.06 | 68.48 |
| Feb. | 11.91 | 78.73 | 75.84 | 14.21 | 44.12 | 70.07 |
| March | 15.55 | 53.19 | 63.02 | 16.03 | 15.95 | 58.65 |
| April | 15.27 | 53.96 | 61.08 | 14.87 | 40.93 | 54.68 |
| May | 17.71 | 17.29 | 44.75 | 17.84 | 9.47 | 38.44 |
| Mean | 16.02 | 34.66 | 63.58 | 16.29 | 25.23 | 59.8 |

Adopted from CAEWRI, National Agricultural Research Centre, Islamabad. Av. Temp—Average temperature; R.H—relative humidity, mm = millimeter.

**Table 2.** Nutrient composition of different composts applied as treatments in the experiments.

| Compost | Av. P (%) | Total N (%) | TOC (%) | C:N |
|---|---|---|---|---|
| SPLC | 0.35 | 1.35 | 19.36 | 14.34 |
| $RPEC_1$ | 1.72 | 1.29 | 16.3 | 12.66 |
| $RPEC_2$ | 1.24 | 1.28 | 17.7 | 13.83 |

SPLC—Simple poultry litter; $RPEC_1$—Poultry litter + rock phosphate + *Pseudomonas* sp.; $RPEC_2$—Poultry litter + rock phosphate + *Proteus* sp.; Av. P—Available phosphorus; N—Nitrogen; TOC—Total organic carbon; C:N; Carbon–nitrogen ratio.

### 2.2. Seed Inoculation

The PGPR strains; *Pseudomonas* sp. (Accession no. KF307201) and *Proteus* sp. (Accession no. KF307202) were used at $6 \times 10^8$ CFU/mL for seed inoculation. Wheat seeds were inoculated with

cultures for 4 h and then the seeds were shade dried before sowing. The inoculants were applied individually as well as in combination with organic and inorganic fertilizer treatments.

### 2.3. Yield, Physiology and Plant Nutrient Analysis

Growth and yield parameters; the number of tillers, grain yield and total dry matter yield were recorded at the time of harvesting. However, for the determination of dry matter yield, the aerial part of the plant from each plot was harvested. Then the spikes were separated from harvested plants of respective treatments and the grains of each pot were weighed to calculate grain yield [kg ha$^{-1}$].

Chlorophyll and phytohormones (IAA, GA) were analyzed in flag leaves of the wheat plants. Chlorophyll was recorded by using SPAD chlorophyll meter [Konica Minolta, Langenhagen, Germany], while leaf IAA and GA were extracted through the method of Kettner and Doerffling [28] and analyzed on HPLC (Agilent 1100, Waldbronn, Germany) using UV detector and C-18 column (39 × 300 mm). Methanol, acetic acid, and water (30:1:70) were used as mobile phase. The wavelength used for the detection of IAA was 280 nm [29] whereas for GA, it was adjusted at 254 nm. These hormones were identified on the basis of retention time and peak area of the standards. Pure IAA and GA$_3$ (Sigma Chemicals Co. Ltd. St. Louis, Missouri, USA) were used as standard for identification and quantification of plant hormones. The above ground plants were harvested from each plot, dried at 70 °C for 48 h, ground at the grinding mill and samples were stored in Ziploc polyethylene bags at room temperature till nutrient analysis. Total phosphorus in plant samples and in seeds was analyzed through Olsen and Sommers [30]. However, phosphorus concentration in shoot was used for the calculation of plant P uptake (kg ha$^{-1}$).

### 2.4. Soil Analysis

Initial soil samples were taken for physicochemical properties (Table 3). Soil samples (0–30 cm) were analyzed for the texture [31], organic matter [32], total P [33]. However, soil samples were extracted through Ammonium Bicarbonate Diphenyl Triamine Penta Acetic Acid (AB-DTPA) solution for determination of available P, NO$_3$-N and extractable K following the method of Soltanpour and Schwab [34] and soil pH (1:5 soil–water) using the method of Mclean [35]. Undisturbed soil samples were collected for soil bulk density (g cm$^{-3}$) using stainless steel cylinders [36]. Soil phosphatase activity was determined by the method of Tabatabai and Bremner [37], whereas, microbial biomass carbon and microbial biomass phosphorus was determined following the method adopted by Steel and Torriej [38]. For determining post-harvest soil properties, soil samples were collected after 6 days of wheat crop harvesting.

**Table 3.** Physicochemical, biological and biochemical properties of soil.

| Properties | 2010–2011 | 2011–2012 |
|---|---|---|
| Texture | Silty Loam | Silty Loam |
| Sand | 18% | 18% |
| Silt | 52% | 50% |
| Clay | 30% | 32% |
| pH | 7.48 | 7.5 |
| Ec (dSm$^{-1}$) | 0.45 | 0.46 |
| Bulk density (g cm$^{-3}$) | 1.43 | 1.42 |
| NO$_3$-N (mg kg$^{-1}$) | 3.23 | 3.4 |
| Total Phosphorus (mg kg$^{-1}$) | 500 | 482 |
| Available Phosphorus (mg kg$^{-1}$) | 2.7 | 2.8 |
| Extractable potassium (mg kg$^{-1}$) | 96 | 92 |

**Table 3.** *Cont.*

| Properties | 2010–2011 | 2011–2012 |
|---|---|---|
| Organic matter (%) | 0.86 | 0.84 |
| Cu ($\mu$g g$^{-1}$) | 1.1 | 0.78 |
| Fe ($\mu$g g$^{-1}$) | 54.38 | 58.98 |
| Zn ($\mu$g g$^{-1}$) | 1.7 | 1.64 |
| Mn ($\mu$g g$^{-1}$) | 1 | 1.32 |
| Microbial Biomass carbon (mg kg$^{-1}$) | 83 | 84 |
| Microbial Biomass phosphorus (mg kg$^{-1}$) | 7 | 8 |
| Alkaline phosphatase activity ($\mu$g g$^{-1}$) | 110 | 112 |

## 2.5. Statistical Analysis

The experiment was laid down following the randomized complete block design (RCBD) with split plot design. Different soil amendments (composts and inorganic fertilizers) were assigned with the main plot while PGPRs were placed in sub-plots of the field. Analysis of variance (ANOVA) was conducted with the General Linear Models and means were compared according to the Tukey HSD test with Statistix 8.1 [39]. Two years of data were pooled because there were not interactions between the two years and year was included as a random effect in statistical model.

## 3. Results

Means of two-year data (2010–2011 and 2011–2012) are provided here due to the result similarity trend from both the years.

### 3.1. Yield and Yield Components

The data for the number of tillers showed 36%, 34%, 30%, 24% and 21% increases with un-inoculated RPEC$_1$, FDP, RPEC$_2$, SPLC and HDP, respectively, over un-inoculated untreated control (Table 4). The treatment RP did not show any significant increase over un-inoculated untreated control. Seed inoculation with PGPRs without any fertilizer treatment did not show any difference with un-inoculated control. However, seed inoculation with *Pseudomonas* sp. in combination with RPEC$_1$ treatment showed a maximum 5% increase in the number of tillers over un-inoculated RPEC$_1$ and FDP.

C—Control (un-inoculated untreated), SPLC—Simple poultry compost, RPEC$_1$—Rock phosphate enriched compost inoculated with *Pseudomonas* species, RPEC$_2$—Rock phosphate enriched compost inoculated with *Proteus* species, RP—Rock phosphate, HDP—Half dose inorganic P fertilizer, FDP—Full dose inorganic P fertilizer.

The data presented in Table 4, showed that there was a significant ($p \leq 0.05$) effect of PGPR on grain yield of wheat crop. A maximum (18%) increase in grain yield was recorded in plants inoculated with *Pseudomonas* sp. which was 4% higher than inoculation with *Proteus* sp. Without inoculation, maximum (67%) increase in grain yield was recorded with the application of RPEC$_1$ which was 4%, 9% and 16% higher than FDP, RPEC$_2$ and SPLC, respectively over un-inoculated untreated control. However, RPEC$_2$ showed a 52% increase in grain yield over control. The interactive effect of fertilizers × PGPR, was highly significant ($p \leq 0.05$) for grain yield. *Pseudomonas* sp. inoculated RPEC$_1$ and FDP gave maximum (10%) increase over un-inoculated RPEC$_1$ and FDP treatments, respectively. The *Proteus* sp. in combination with RPEC$_1$ also showed 3% increase over un-inoculated RPEC$_1$ treatment, whereas the treatment RP produced minimum grain yield showing 14% increase over un-inoculated RP treatment.

The data in Table 4 show that the treatment RPEC$_1$ resulted in a maximum increase in dry matter yield which was 3.8%, 16%, 27% higher than FDP, RPEC$_2$ and SPLC respectively, over un-inoculated untreated control. The stimulatory effect of PGPR was recorded on dry matter yield. However, the interactive effect of PGPR and fertilizer treatments was significant for dry matter yield of wheat crop. The combination of *Pseudomonas* sp. with RPEC$_1$ gave the maximum increase (62%) similar to

FDP (60%) while with *Proteus* sp. in combination with RPEC$_1$ showed 56% increase over un-inoculated untreated control. RP inoculation with *Proteus* sp. showed nonsignificant difference with un-inoculated untreated control.

**Table 4.** Effects of plant growth promoting rhizobacteria (PGPR), P-enriched compost and inorganic fertilizers on yield and yield components on wheat.

| Treatments | C | SPLC | RPEC$_1$ | RPEC$_2$ | RP | HDP | FDP |
|---|---|---|---|---|---|---|---|
| **Number of Tillers (m$^{-2}$)** | | | | | | | |
| Without inoculation | 260 [e] (±2.22) | 323 [c] (±1.42) | 354 [a] (±2.83) | 339 [b] (±3.53) | 261 [e] (±3.12) | 315 [c,d] (±3.9) | 347 [ab] (±2.88) |
| *Proteus* sp. | 263 [e] (±2.98) | 331 [c] (±2.17) | 361 [a] (±2.67) | 347 [b] (±4.94) | 264 [e] (± 4.24) | 321 [d] (±3.87) | 352 [ab] (±4.72) |
| *Pseudomonas* sp. | 267 [f] (±3.61) | 343 [d] (±3.14) | 372 [a] (±4.88) | 356 [b,c] (±5.01) | 269 [f] (±3.9) | 326 [e] (±2.84) | 364 [ab] (±2.92) |
| **Grain yield (kg ha$^{-1}$)** | | | | | | | |
| Without inoculation | 2177 [f,g] (±29.22) | 3120 [d] (±36.09) | 3629 [a] (±210.74) | 3317 [c] (±24.36) | 2195 [f] (±29.032) | 3020 [d,e] (±25.78) | 3495 [b] (±31.16) |
| *Proteus* sp. | 2474 [f,g] (±20) | 3319 [e] (±25.45) | 3731 [a] (±33.22) | 3524 [c] (±24.67) | 2498 [f] (±36.103) | 3511 [c,d] (±23.24) | 3620 [b] (±29.2) |
| *Pseudomonas* sp. | 2571 [f,j] (±32.89) | 3433 [d] (±34.36) | 3987 [a] (±39.84) | 3639 [c] (±23.48) | 2592 [f] (±22.86) | 3364 [d,e] (±38.96) | 3848 [b] (±29.07) |
| **Dry matter yield (kg ha$^{-1}$)** | | | | | | | |
| Without inoculation | 8955 [f,g] (±38.43) | 10,804 [d] (±31.9) | 13,714 [a] (±26.7) | 11,773 [c] (±32.33) | 8969 [f] (±24.41) | 10,746 [d,e] (±35.9) | 13,208 [b] (±24.47) |
| *Proteus* sp. | 8978 [e] (±28.97) | 10,930 [c] (±39.93) | 13,926 [a] (±35.55) | 12,077 [c] (±26.23) | 8987 [e] (±34.48) | 10,806 [c,d] (±37.13) | 13,358 [b] (±27.2) |
| *Pseudomonas* sp. | 9005 [f,g] (±34.02) | 11,138 [d] (±0.67) | 14,503 [a] (±37.14) | 12,651 [c] (±36.29) | 9050 [f] (±45) | 10,891 [e] (±34.65) | 14,382 [a,b] (±25.37) |

All the treatments sharing common letter are similar otherwise they differ significantly at $p \leq 0.05$.

## 3.2. Leaf Chlorophyll, IAA and GA Contents

Mean data recorded for chlorophyll contents in flag leaves of wheat crop showed that there was a significant ($p \leq 0.05$) difference for the treatments (Figure 1). Among the un-inoculated treatments, RPEC$_1$ showed the highest (28%) increase which was 2%, 6%, 12% and 25% higher than FDP, RPEC$_2$, SPLC and HDP, respectively. Seed inoculation with *Pseudomonas* sp. resulted in an increase (4%) in chlorophyll content over un-inoculated treatments. However, the interactive effect of treatments (PGPRs × fertilizer) showed 29% increase followed by FDP (27%) over un-inoculated untreated control. While the treatment RP showed a nonsignificant difference when applied in combination with *Pseudomonas* as well as *Proteus* sp.

Data in Figure 2 show that RPEC$_1$ resulted maximum (12%) increase in IAA content, having a similar effect as with FDP, followed by RPEC$_2$ showing a 9% increase, while HDP and SPLC showed a similar effect (i.e., 7% increase) in IAA content over un-inoculated untreated control. The inoculation of seeds with *Pseudomonas* sp. gave higher values of IAA by showing 6% increase over un-inoculated treatments. The interactive effect of PGPR × Fertilizer was nonsignificant except for *Pseudomonas* sp. which showed a 20% increase, when used in combination with RPEC$_1$ and FDP treatments.

The treatment RPEC$_1$ resulted in a 13% increase in GA content, followed by FDP (11%), RPEC$_2$ (9%), SPLC (6%), while HDP resulted only a 4% increase over un-inoculated untreated control (Figure 3). However, seed inoculation with *Pseudomonas* sp. showed a maximum (5%) increase over un-inoculated RPEC$_2$ treatment. The data showed that the treatments RPEC$_1$ and FDP in combination with *Pseudomonas* sp. showed a maximum (16%) increase in GA contents of flag leaves. PGPR inoculation with RP and HDP showed a nonsignificant difference among respective un-inoculated treatments.

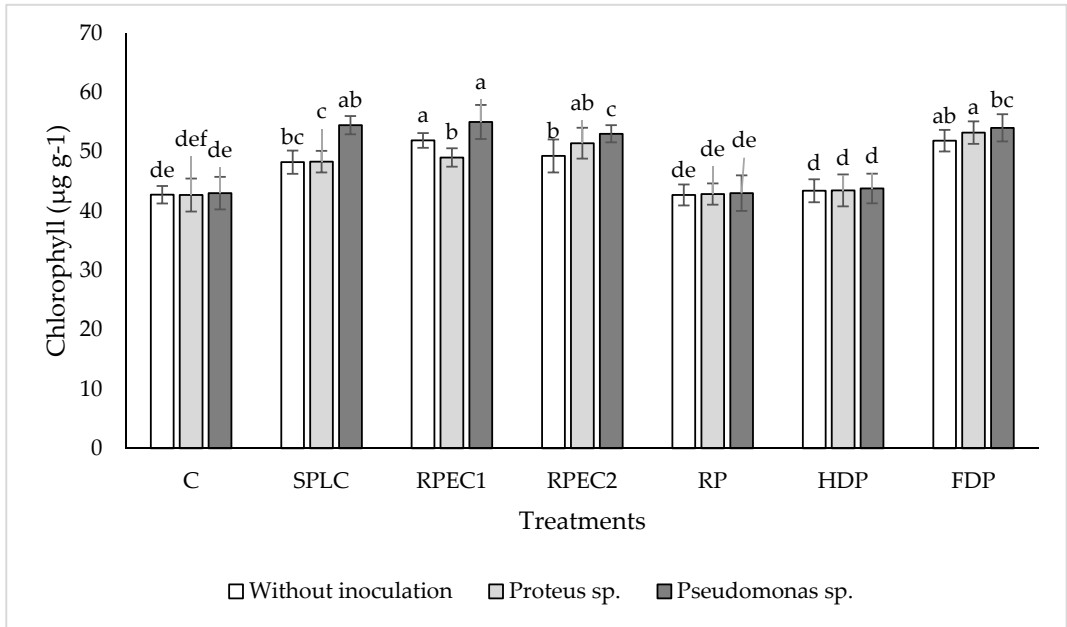

**Figure 1.** Effects of PGPR, P-enriched compost and inorganic fertilizers on leaf chlorophyll concentration ($\mu g\ g^{-1}$). C—Control; SPLC—Poultry litter only; RPEC1—Rock phosphate + poultry litter solubilized with *Pseudomonas* sp. during the composting process; RPEC$_2$—Rock phosphate + poultry litter solubilized with *Proteus* sp. during composting process), RP—Rock phosphate + poultry litter; HDP—Half dose inorganic P from Single Super Phosphate-SSP 18% P$_2$O$_5$; FDP—Chemical fertilizer (Single Super Phosphate). All the treatments sharing a common letter are similar, otherwise they differ significantly at $p \leq 0.05$.

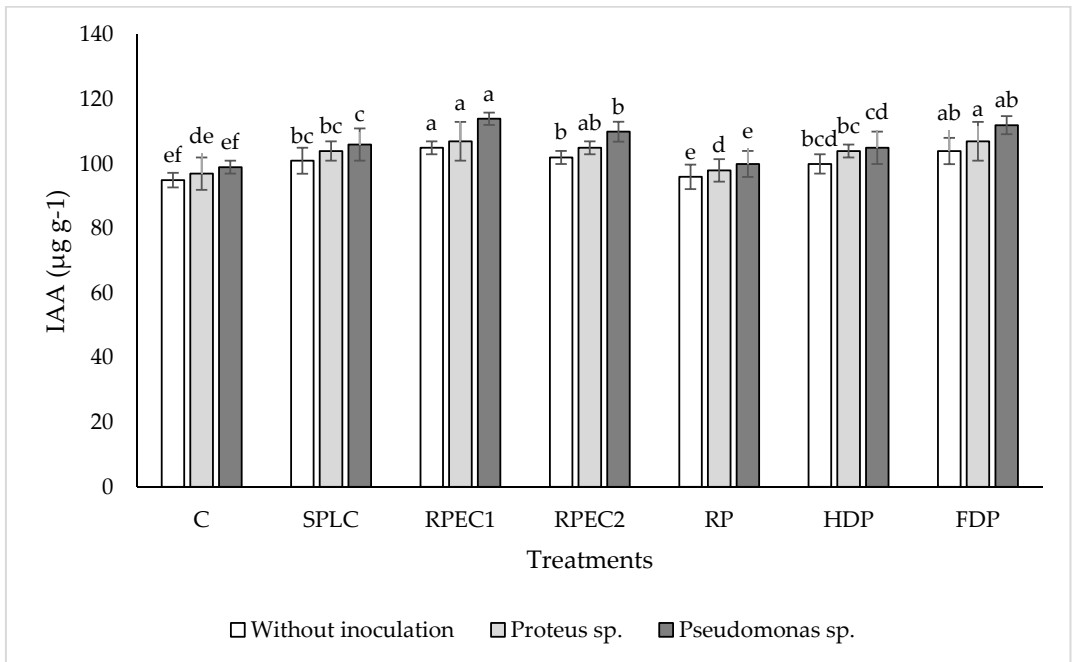

**Figure 2.** Effects of PGPR, P-enriched compost and inorganic fertilizers on leaf IAA concentration ($\mu g\ g^{-1}$) in wheat. C—Control; SPLC—Poultry litter only; RPEC1—Rock phosphate + poultry litter solubilized with *Pseudomonas* sp. during the composting process; RPEC$_2$—Rock phosphate + poultry litter solubilized with *Proteus* sp. during composting process), RP—Rock phosphate + poultry litter; HDP—Half dose inorganic P from Single Super Phosphate—SSP 18% P$_2$O$_5$; FDP- Chemical fertilizer (Single Super Phosphate). All the treatments sharing common letter are similar otherwise they differ significantly at $p \leq 0.05$.

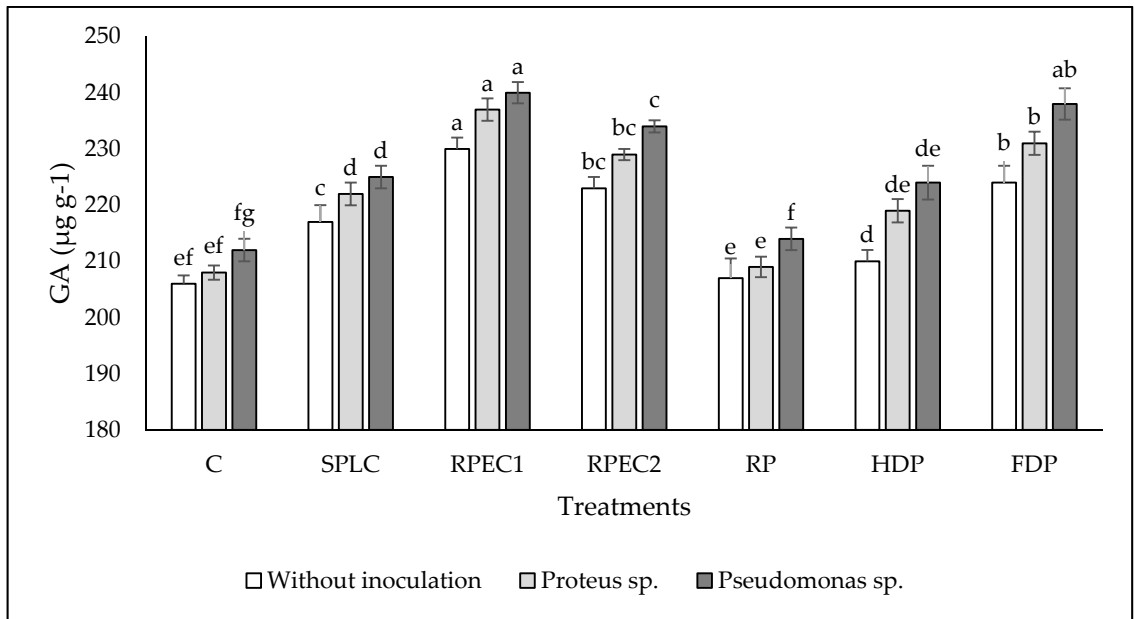

**Figure 3.** Effects of PGPR, P-enriched compost and inorganic fertilizers on leaf GA concentration ($\mu g\ g^{-1}$) in wheat. C—Control; SPLC—Poultry litter only; RPEC1—Rock phosphate + poultry litter solubilized with *Pseudomonas* sp. during the composting process; $RPEC_2$—Rock phosphate + poultry litter solubilized with *Proteus* sp. during composting process), RP—Rock phosphate + poultry litter; HDP—Half dose inorganic P from Single Super Phosphate—SSP 18% $P_2O_5$; FDP—Chemical fertilizer (Single Super Phosphate). All the treatments sharing common letter are similar otherwise they differ significantly at $p \leq 0.05$.

### 3.3. Plant Phosphorus Uptake and Seed Phosphorus

The data presented in Figure 4 show that the phosphorus uptake was maximum (70%) due to the application of $RPEC_1$ followed by $RPEC_2$ (63%) and FDP (60%), while RP treatment showed no significant difference compared to un-inoculated untreated control. Seed inoculation with *Pseudomonas* sp. resulted in a maximum (7%) increase in P-uptake over un-inoculated treatments. The interaction of fertilizer treatments and PGPRs showed that $RPEC_1$ in combination with *Pseudomonas* sp. showed maximum increase (88%) in P-uptake followed by *Proteus* sp. inoculated $RPEC_1$ (79%) over untreated un-inoculated control.

The seed phosphorus content showed a 61% increase following application of $RPEC_1$ over un-inoculated untreated control, which was 12%, 17%, 33% and 41% higher than FDP, $RPEC_2$, SPLC and HDP, respectively (Figure 5). The application of *Pseudomonas* sp. alone also resulted in an increase (3.5%) in seed P contents over un-inoculated untreated control whereas the interactive effect of PGPR × Fertilizer was nonsignificant.

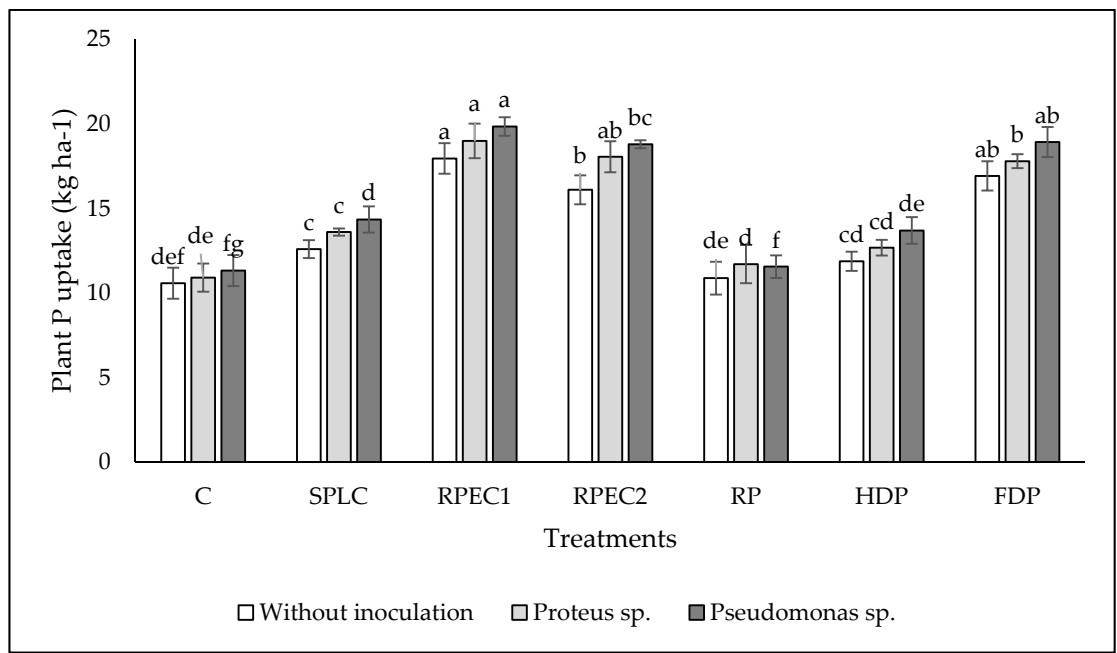

**Figure 4.** Effects of PGPR, P-enriched compost and inorganic fertilizers on plant P uptake (kg ha$^{-1}$) in wheat. C—Control; SPLC—Poultry litter only; RPEC1—Rock phosphate + poultry litter solubilized with *Pseudomonas* sp. during the composting process; RPEC$_2$—Rock phosphate + poultry litter solubilized with *Proteus* sp. during composting process), RP—Rock phosphate + poultry litter; HDP— Half dose inorganic P from Single Super Phosphate—SSP 18% P$_2$O$_5$; FDP—Chemical fertilizer (Single Super Phosphate). All the treatments sharing common letter are similar otherwise they differ significantly at $p \leq 0.05$.

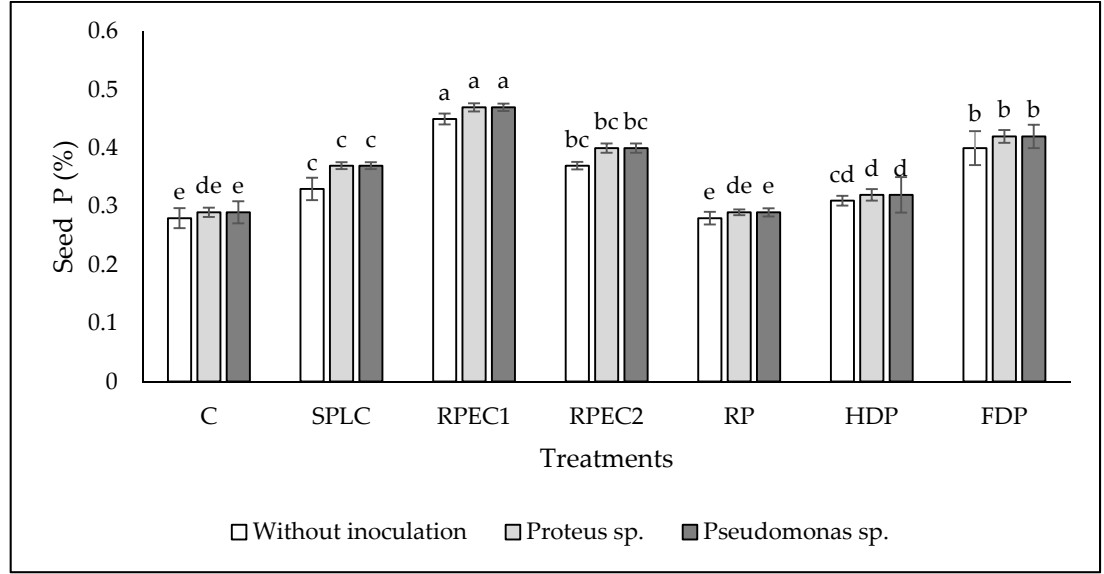

**Figure 5.** Effects of PGPR, P-enriched compost and inorganic fertilizers on plant P uptake (kg ha$^{-1}$) in wheat. C—Control; SPLC—Poultry litter only; RPEC1—Rock phosphate + poultry litter solubilized with *Pseudomonas* sp. during the composting process; RPEC$_2$—Rock phosphate + poultry litter solubilized with *Proteus* sp. during composting process), RP—Rock phosphate + poultry litter; HDP— Half dose inorganic P from Single Super Phosphate—SSP 18% P$_2$O$_5$; FDP—Chemical fertilizer (Single Super Phosphate). All the treatments sharing common letter are similar otherwise they differ significantly at $p \leq 0.05$.

### 3.4. Soil Properties

3.4.1. Available P, Nitrate Nitrogen and Extractable Potassium

The post-harvest soil analysis for phosphorus availability showed that the treatments significantly increased the P availability (Table 5). The treatment $RPEC_1$ resulted in a significant increase over un-inoculated untreated control, the value of which was 37%, 82%, and 130% higher than SPLC, HDP and RP, respectively. The PGPR seed inoculation effect was significant ($p \leq 0.05$) for post-harvest available soil P contents. The phosphorus content was increased (20% and 9%) in the rhizosphere of plants treated with *Pseudomonas* sp. and *Proteus* sp. respectively. In combination with *Pseudomonas* sp., the treatment $RPEC_1$ gave maximum increase (3.43-fold) over un-inoculated untreated control, while $RPEC_1$ in combination with *Proteus* sp. and *Pseudomonas* sp. in combination with FDP showed similar results by giving a 3.17-fold increase over the un-inoculated untreated control. The PGPRs (*Pseudomonas* sp. and *Proteus* sp.) in combination with $RPEC_2$ showed a similar effect for increase [8%] in available P over un-inoculated $RPEC_2$. The treatment RP in combination with *Pseudomonas* sp. resulted in 30% increase over un-inoculated RP, which was 16.5% higher than *Proteus* sp. inoculated RP.

C—Control (un-inoculated untreated), SPLC—Simple poultry compost, $RPEC_1$—Rock phosphate enriched compost inoculated with *Pseudomonas* species, $RPEC_2$—Rock phosphate enriched compost inoculated with *Proteus* species, RP—Rock phosphate, HDP—Half dose inorganic P fertilizer, FDP—Full dose inorganic P fertilizer. Soil samples for nutrient and biological analyses were collected two days after wheat harvesting.

**Table 5.** Effects of PGPR, P-enriched compost and inorganic fertilizers on post-harvest soil of wheat in field experiments.

| Treatments | C | SPLC | $RPEC_1$ | $RPEC_2$ | RP | HDP | FDP |
|---|---|---|---|---|---|---|---|
| **Available P (mg kg$^{-1}$)** | | | | | | | |
| Without inoculation | 2.8 [g] | 6.0 [e] | 10.9 [a] | 9.7 [b] | 4.4 [f] | 7.8 [d] | 8.76 [c] |
| *Proteus* sp. | 2.9 [g] | 6.3 [e] | 11.6 [a] | 10.1 [b] | 4.9 [f] | 8.4 [c,d] | 8.90 [c] |
| *Pseudomonas* sp. | 3.2 [g] | 6.9 [e] | 14.4 [a] | 10.48 [b] | 5.7 [f] | 9.2 [c,d] | 9.74 [b,c] |
| **Nitrate nitrogen (mg kg$^{-1}$)** | | | | | | | |
| Without inoculation | 3.07 [f] | 3.87 [a,b,c] | 3.98 [a] | 3.92 [a,b] | 3.07 [f] | 3.39 [e] | 3.78 [b,c,d] |
| *Proteus* sp. | 3.08 [e] | 3.88 [a,b] | 4.04 [a] | 4.07 [a] | 3.07 [e] | 3.42 [d] | 3.81 [b,c] |
| *Pseudomonas* sp. | 3.0 [f] | 3.89 [b,c] | 4.20 [a] | 4.14 [a,b] | 3.08 [f] | 3.44 [e] | 3.85 [c,d] |
| **Extractable potassium (mg kg$^{-1}$)** | | | | | | | |
| Without inoculation | 95.50 [c,d] | 106 [a,b] | 109.5 [a] | 106 [a,b] | 96.5 [c,d] | 101 [a,b,c] | 102 [a,b,c] |
| *Proteus* sp. | 96.33 [d] | 108 [a,b] | 110 [a] | 107 [a,b] | 97 [c,d] | 102 [a,b,c] | 103 [a,b,c] |
| *Pseudomonas* sp. | 97.2 [d] | 108 [a,b] | 112 [a] | 109 [a,b] | 98 [c,d] | 104 [b,c] | 104 [b,c] |
| **Alkaline phosphatase (µg PNP g$^{-1}$ hr$^{-1}$)** | | | | | | | |
| Without inoculation | 117 [d] | 136 [b] | 151 [a] | 143 [a,b] | 118 [d] | 127 [c] | 133 [b,c] |
| *Proteus* sp. | 118 [f,g] | 142 [b,c] | 157 [a] | 147 [b] | 121 [f] | 132 [d,e] | 137.5 [c,d] |
| *Pseudomonas* sp. | 120 [e] | 155.5 [b] | 167 [a] | 156 [b] | 122 [e] | 136 [c,d] | 143.5 [c] |
| **Microbial biomass carbon (µg g$^{-1}$)** | | | | | | | |
| Without inoculation | 84 [f,g] | 112 [c] | 136 [a] | 123 [b] | 87 [e,f] | 93 [e] | 106 [c,d] |
| *Proteus* sp. | 89 [f,g] | 119 [c] | 144 [a] | 137 [a,b] | 92 [f] | 103 [e] | 114 [c,d] |
| *Pseudomonas* sp. | 93 [f,g] | 127 [c] | 159 [a] | 146 [b] | 96 [f] | 114 [d,e] | 120 [c,d] |
| **Microbial biomass phosphorus (µg g$^{-1}$)** | | | | | | | |
| Without inoculation | 7 [d] | 14 [b] | 19 [a] | 18 [a] | 7 [d] | 10 [b,c] | 11 [b,c] |
| *Proteus* sp. | 8 [d] | 16 [b,c] | 21 [a] | 19 [a,b] | 8 [d] | 11 [c,d] | 12 [c,d] |
| *Pseudomonas* sp. | 8 [e] | 18 [c] | 26 [a] | 24 [a,b] | 8 [e] | 13 [d] | 13 [d] |

All the treatments sharing a common letter are similar otherwise they differ significantly at $p \leq 0.05$.

Mean data for post-harvest soil nitrate-nitrogen showed significant differences with the application of fertilizer treatments (Table 5). The treatments $RPEC_1$ and $RPEC_2$ resulted a 36% increase followed by SPLC, FDP and HDP showing 29%, 26% and 14% increase over un-inoculated untreated control, respectively. The treatment RP showed nonsignificant difference with the control. There was a nonsignificant effect of seed inoculation on $NO_3$-N over un-inoculated control. However, the maximum increase was recorded by the application of *Pseudomonas* sp. which was 4% higher over un-inoculated control treatments. The interactive effect of PGPR and fertilizer treatments was nonsignificant with SPLC, RP, HDP and FDP, while the treatments $RPEC_1$ and $RPEC_2$ showed 36% and 35% increases over un-inoculated untreated control. The mean data showed that all the treatments increased extractable potassium except RP (Table 5). A maximum increase (15%) in the content of extractable K was recorded following the treatment $RPEC_1$ followed by SPLC and $RPEC_2$ which were significantly similar in their effect by showing 11% and 12% increase over un-inoculated untreated control, respectively. The treatments FDP and HDP also showed similar effects and increased extractable K by 7% over control. Seed inoculation with *Pseudomonas* sp. showed 2% increase in seed phosphorus contents over un-inoculated untreated control. The interactive effect of PGPR × Fertilizers, was nonsignificant, whereas *Pseudomonas* sp. inoculation in combination with $RPEC_1$ showed maximum (17%) phosphorus contents over un-inoculated untreated control.

### 3.4.2. Alkaline Phosphatase and Microbial Biomass

Alkaline phosphatase activity was significantly ($p \leq 0.05$) increased as a result of different treatments (Table 5). The treatment $RPEC_1$ resulted a maximum increase (29%) over un-inoculated untreated control, which was 5.6%, 11%, 13.5% and 19% higher than $RPEC_2$, SPLC, FDP and HDP respectively. *Pseudomonas* sp. inoculation showed 8% increase over un-inoculated treatments, which was 2.5% higher than *Proteus* sp. inoculation. Significant ($p \leq 0.05$) increase in alkaline phosphatase activity was recorded due to the combine effects of PGPRs with different fertilizer treatments. The inoculation of *Pseudomonas* sp. in combination with the treatment $RPEC_1$ showed maximum (43%) increase over untreated un-inoculated control. The treatments $RPEC_2$ and SPLC in combination with *Pseudomonas* sp. and $RPEC_1$ in combination with *Proteus* sp. showed similar effect and increased the alkaline phosphatase activity by 34% over un-inoculated untreated control. The treatment FDP in combination with *Pseudomonas* sp. showed a 23% increase over untreated un-inoculated control; the effect of which was significantly similar to un-inoculated $RPEC_2$ treatment. The PGPRs (*Pseudomonas* sp. and *Proteus* sp.) in combination with HDP showed a similar effect but significantly lower percentage increase than un-inoculated HDP treatment.

Mean data showed that the fertilizer treatments significantly ($p \leq 0.05$) improved the microbial biomass carbon contents (Table 5). Significant increase (65%) was recorded in microbial biomass carbon contents in $RPEC_1$ treatment over control; the increase for $RPEC_1$ was also 9%, 22%, 30%, 43%, and 60% higher than that of $RPEC_2$, SPLC, FDP, HDP and RP, respectively. Inoculation of seeds with *Pseudomonas* sp. showed maximum (16%) increase in microbial biomass carbon over un-inoculated treatments, the values of which were 8% higher than *Proteus* sp. inoculated treatments. The interactive effect of PGPR inoculation to seeds and fertilizer treatments was also significant with microbial biomass carbon contents. Among the *Pseudomonas* sp. inoculated treatments, $RPEC_1$ showed 89% increase in microbial biomass carbon over un-inoculated untreated control, while the treatment $RPEC_2$ showed 74% increase, which was significantly similar with *Proteus* sp. inoculated $RPEC_1$ treatment. However, *Proteus* sp. inoculated $RPEC_2$ increased microbial biomass carbon by 63% and showed a nonsignificant difference with un-inoculated $RPEC_1$ treatment. *Pseudomonas* sp. in combination with FDP showed a nonsignificant difference with un-inoculated $RPEC_2$ but was 13% higher than the un-inoculated FDP treatment. The treatment RP in combination with *Pseudomonas* sp. increased microbial biomass by 10% over untreated un-inoculated control.

The data in Table 5 show that microbial biomass phosphorus (MBP) increased significantly ($p \leq 0.05$) with the application of fertilizer treatments than the un-inoculated untreated control. The maximum increase (1.75-fold) in MBP was recorded from the treatment $RPEC_1$ which was 7%, 37% and 83%

higher than RPEC$_2$, SPLC and FDP, respectively while FDP showed nonsignificant difference with HDP. The treatment RP showed a nonsignificant difference with un-inoculated untreated control. There was also a significant effect of PGPR on microbial biomass phosphorus. *Pseudomonas* sp. inoculation increased microbial biomass P by 33% over un-inoculated treatments, the values of which were 14% higher than *Proteus* sp. inoculated treatments. *Pseudomonas* sp. inoculation with RPEC$_1$ showed a maximum increase (2.7-fold) in MBP followed by RPEC$_2$ (2.43-fold) over un-inoculated untreated control. The treatment RPEC$_1$ in combination with *Proteus* sp. showed an (61%) increase over un-inoculated untreated control which was at par with *Pseudomonas* sp. inoculated SPLC. The treatments FDP, HDP increased (46%) microbial biomass P showing nonsignificant difference with each other and RP showed 12% increase in microbial biomass P with PGPR inoculation over un-inoculated untreated control.

### 3.5. Economic Analysis

The economic analysis of applied treatments (Table 6) in terms of value cost ratio (VCR) showed that RPEC$_1$ performed best with and without seed inoculation. Among the un-inoculated treatments, RPEC$_1$ showed maximum VCR (2.72) followed by RPEC$_2$ (2.14), FDP (1.94) while the minimum (0.06) VCR was received from RP. Seed inoculation with *Pseudomonas* sp. in combination with RPEC$_1$ superseded all of the treatments resulting in maximum VCR (3.23). Hence, rock phosphate enriched compost alone or more so in combination with phosphate solubilizing bacteria (PSB), can perform better than chemical fertilizers. The economic analysis revealed that RPEC could be an economically feasible substitute to costly chemical fertilizers for sustainable crop production.

**Table 6.** Economic Analysis of the applied products presented as value cost ratio (VCR).

| Treatments | Grain Yield | Increase in Yield | Increased Yield Value | Cost of Inputs | Net Return | VCR |
|---|---|---|---|---|---|---|
| | kg ha$^{-1}$ | | Rs. ha$^{-1}$ | | | |
| **Control** | **2177** | - | - | - | - | - |
| SPLC | 3120 | 943 | 28,290 | 17,760 | 10,530 | 1.59:1 |
| RPEC$_1$ | 3629 | 1452 | 43,560 | 16,010 | 27,550 | 2.72:1 |
| RPEC$_2$ | 3317 | 1140 | 34,200 | 16,010 | 18,190 | 2.14:1 |
| RP | 2195 | 18 | 540 | 9285 | −8745 | 0.06:1 |
| HDP | 3020 | 843 | 25,290 | 13,435 | 11,855 | 1.88:1 |
| FDP | 3495 | 1318 | 39,540 | 20,385 | 19,155 | 1.94:1 |
| *Proteus* sp. (1S) | 2474 | 297 | 8910 | 7310 | 1600 | 1.22:1 |
| SPLC + 1S | 3319 | 1142 | 34,260 | 18,560 | 15,700 | 1.85:1 |
| RPEC$_1$+1S | 3731 | 1554 | 46,620 | 16,810 | 29,810 | 2.77:1 |
| RPEC$_2$+1S | 3524 | 1347 | 40,410 | 16,810 | 23,600 | 2.40:1 |
| RP+1S | 2498 | 321 | 9630 | 10,085 | −455 | 0.95:1 |
| HDP+1S | 3511 | 1334 | 40,020 | 14,235 | 25,785 | 2.01:1 |
| FDP+1S | 3620 | 1443 | 43,290 | 21,185 | 22,105 | 2.04:1 |
| *Pseudomonas* (2S) | 2571 | 394 | 11,820 | 7310 | 4510 | 1.62:1 |
| SPLC + 2S | 3433 | 1256 | 37,680 | 18,560 | 19,120 | 2.03:1 |
| RPEC$_1$+ 2S | 3987 | 1810 | 54,300 | 16,810 | 37,490 | 3.23:1 |
| RPEC$_2$+ 2S | 3639 | 1462 | 43,860 | 16,810 | 27,050 | 2.61:1 |
| RP+ 2S | 2592 | 415 | 12,450 | 10,085 | 2365 | 1.23:1 |
| HDP+ 2S | 3364 | 1187 | 35,610 | 14,235 | 21,375 | 2.05:1 |
| FDP+ 2S | 3848 | 1671 | 50,130 | 21,185 | 28,945 | 2.37:1 |

Increase in yield = Yield of treatment − Yield of control, Increased yield value = Grain price × increase in yield, Net return = Increased yield value − cost of inputs, Value cost ratio (VCR) = Increased yield value/cost of inputs, Poultry litter = Rs. 1.5 kg$^{-1}$, Rock phosphate = Rs. 5 kg$^{-1}$, Single super phosphate (SSP) = Rs. 25 kg$^{-1}$, Urea = Rs. 30 kg$^{-1}$, Labor charges for compost preparation = Rs. 5250, Seed inoculant = Rs. 200 L$^{-1}$, Wheat grain price = Rs. 30 kg$^{-1}$, SPLC—Simple Poultry litter, RPEC$_1$—Rock Phosphate Enriched Compost solubilized with *Pseudomonas* sp., RPEC$_2$—Rock phosphate enriched compost solubilized with *Proteus* sp. 1S-Seed inoculation with *Proteus* sp., 2S—Seed inoculation with *Pseudomonas* sp., RP—Rock phosphate, HDP—Half dose of inorganic fertilizer, FDP—Full dose of inorganic fertilizer. Rs—Refer to national currency (Rupees).

## 4. Discussion

The use of plant growth promoting rhizobacteria (PGPR) in combination with organic (composts, rock phosphate) and inorganic (chemical fertilizers) phosphorus sources significantly increased the number of tillers per plant and yield components of wheat crop. The results are in conformity to the findings of Akhtar et al. [40] who recorded increase in plant height, the number of tillers, grain yield and 1000 grain weight of wheat with the use of compost and PGPR inoculation.

Maximum grain yield was obtained by the application of $RPEC_1$ which was higher than the full dose of inorganic P fertilizers (FDP), irrespective of the PGPR seed inoculants. The observed yield increase from $RPEC_1$ was indicative of the high P availability and greater photosynthesis as observed by an increase in chlorophyll content and dry matter production, which was maximum in $RPEC_1$ over other treatments. Plant P availability as the key factor for maximum plant growth and higher crop production [41]. Although a full dose of P (FDP) as inorganic fertilizer (SSP) is a source of readily available phosphorus necessary for early growth of the plants, at the site of SSP application, production of the least soluble Ca-P compounds due to surface adsorption and precipitation, reduce P availability [10]. The organic acids produced due to compost might have reduced P exchange sites through chelation and released more soluble forms of plant available P [42] compared to SSP which could help increase growth and yield of wheat. Seed inoculation with *Pseudomonas* sp. increased the grain yield with fertilizer treatments, however the maximum increase was recorded from *Pseudomonas* sp. inoculation with $RPEC_1$ followed by the inoculated FDP treatment. Microbial community in the root rhizosphere might have taken part to release fixed phosphorus through organic acids production which ultimately increased the yield of wheat. Afzal and Bano [43] reported that seed inoculation with PGPR in combination with P fertilizer increased the grain yield of wheat which was 30–40% higher than the un-inoculated P fertilizer. It was reported that organic manures and bio-fertilizers have a high impact on nutrient uptake, physiological process of wheat, and also on water holding capacity of the soil which ultimately increase grain yield of the crop [44]. Amujoyegbe et al. [45] recorded higher grain yield of maize due to the application of chicken manure in combination with microbes compared to chemical fertilizer and chicken manure alone. An association of agronomic traits with grain yield and a positive correlation of 1000 grain weight with grain yield was previously demonstrated in PGPR + manure treated plants of wheat [44].

Increase in the dry matter yield due to the application of $RPEC_1$ compared to FDP may be due to higher vegetative growth, chlorophyll content and the maximum number of tillers during the crop growth, while mobilization of phosphorus due to dissolution of rock phosphate from $RPEC_1$ might have taken part in the physiological processes leading to maximum biomass yield. Higher yields of mung-bean were recorded due to bio-inoculated RP enriched compost having higher citrate soluble, water soluble P and organic P, maximum microbial biomass carbon and acid phosphatase activity compared to un-inoculated composts [7]. Similarly, Nishanth and Biswas [11] prepared enriched composts with *Aspergillus awamori* inoculation and tested these on the wheat crop, which gave maximum biomass production in comparison to composts prepared without inoculants. Hossain et al. [46] reported an increase in grain and straw yield of wheat crop with the application of phosphate solubilizing bacteria (PSB) along with different levels of phosphorus. In concurrence with the present results, an increase in dry matter and grain yield of agronomic crops due to phosphate solubilizing microorganisms in combination with different P fertilizers were reported earlier by different workers [47–49].

Phosphorus plays an important role in chlorophyll production and regulation. It has been reported that the partitioning of photosynthates between leaves and reproductive organs is regulated by the availability of phosphorus to the plants [50]. Maximum increase in chlorophyll contents in flag leaves were recorded due to the application of $RPEC_1$ followed by FDP and SPLC. Zafar et al. [51] reported an increase in chlorophyll contents by 10–89% over control in leaves of maize crop following application of P fertilizers in the form of compost and inorganic fertilizers. The PGPR in combination with compost was recorded to be stimulatory for chlorophyll production; this was confirmed for

*Pseudomonas* sp. in combination with $RPEC_1$. Seed inoculation with *Pseudomonas* sp. alone or in combination with P fertilizers, was more efficient for improving chlorophyll contents in flag leaves of wheat plants. Naseem and Bano [52] reported that the seed inoculation with *Pseudomonas* sp. and *Bacillus cereus* increased chlorophyll contents by 8–13% in leaves of wheat crop. An increase in chlorophyll contents with the application of organic manure was also recorded [53].

PGPR alone or in combination with fertilizers showed a significant effect on IAA and GA contents of wheat flag leaves, however, maximum increase was recorded as a result of $RPEC_1$ application followed by FDP and $RPEC_2$. Among the PGPRs, *Pseudomonas* sp. performed better than the *Proteus* sp. Indole Acetic Acid synthesis by bacteria may have various regulatory effects in plant–bacterial interactions and significant effect on plant growth promotion [54]. Generally, phytohormones in plants plays an important role in cell division, proliferation, and differentiation, vascular tissue alteration, responses to light and gravity, general root and shoot architecture, seed and tuber germination, organ differentiation, peak predominance, ethylene synthesis, vegetative growth processes, fruit development and aging. These results are in accordance with the findings of Saharan and Nehra [55], who reported that the phytohormone production through PGPR (*Pseudomonas*, *Azotobacter*, *Azospirillum*) may contribute to growth and yield of the crop. IAA acts as a signal molecule for cell expansion, division and differentiation. Higher counts of genus *Pseudomonas* were recorded [56] in winter wheat cultivars and described the developmental phase of wheat crops as a key factor in higher population of the microbes. The GA and IAA were reported to be produced by bacterial strains such as *Bacillus* and *Pseudomonas* [57] and inoculation of wheat with *Pseudomonas* sp. gave maximum increase in growth and yield [58]. Khan et al. [59] found an increase in IAA and GA contents in leaves of wheat inoculated with *Pseudomonas* and *Bacillus* strains. Sivasankari et al. [60] isolated bacterial strains from black gram (*Vigna mungo*) rhizosphere soil and reported maximum IAA production from *Pseudomonas* sp. than *Proteus* sp.

Phosphors uptake increased with the application of RP enriched compost ($RPEC_1$) which would be due to phosphorus in the soluble form. Higher concentration of macronutrients due to the decomposition of organic materials in the soil were recorded [61]. Incorporation of organic materials can enhance phosphorus availability in the soil solution by decreasing P sorption/fixation through chelation [62]. Phosphorus also plays an efficient role in plant photosynthesis, respiration, formation of cell membrane, glycolysis and enzymes activities [63] showing that the growth and development of all crops are dependent upon P availability [64]. The presence of P as an integral part of nucleotides, phospholipids, phosphoproteins, and coenzymes shows its importance for life [65]. An increase in P-uptake due to enriched compost in the present study was due to the maximum available P as well as total organic and readily available carbon. Sharma et al. [66] reported increased N uptake (18–38 kg $ha^{-1}$), P uptake (2.7–6.6 kg $ha^{-1}$), and K uptake by (16–41 kg $ha^{-1}$) in the rice–wheat system when inoculated with *Pseudomonas striata*. It was reported by Nishanth and Biswas [11] that RP enriched compost inoculated with *Aspergillus awamori* can significantly enhance P uptake in wheat crop, which was recorded as 78% more efficient compared to DAP. Ghaderi et al. [67] reported 51%, 29% and 62% release of phosphorus from iron hydroxides by the application of *Pseudomonas putida*, *Pseudomonas fluorescens*, *and Pseudomonas fluorescens*, respectively. Shrivastava [10] reported that inoculation of microbes with P enriched manure show maximum P uptake in mung-bean crop compared to SSP fertilizer. The P-enriched compost in combination with effective microbes (EM) can enhance N and P uptakes of the cowpea crop [14].

Crop growth is regulated by the nutrient supply from organic or chemical fertilizer sources. Organic materials are considered to be the best source for nutrient supply to plants but with slow release until the crop maturity, which may create a delay in crop maturity or cause high nutrients content in the produce [68]. Maximum P concentration in the wheat seeds with the application of $RPEC_1$ might be due to a slow release process resulting in P accumulation in the seeds due to mobility of the phosphorus from soil to plant process. The integrated management of P fertilizers at the root zone can increase the mobility of P from plant roots through physiological adaptive mechanisms [69]. Seed

inoculation with *Pseudomonas* sp. showed an increase in seed phosphorus. According to Son et al. [70] soybean seed P content increased with inoculation of phosphate solubilizing microorganisms.

Organic and inorganic amendments have a great impact on soil properties [71]; however, while the application of fertilizer increases P availability at all crop growth stages compared to control treatment, the RP compost showed maximum P availability at later stages of wheat crop growth [11]. The increase in post-harvest soil P availability with the application of RP enriched compost may be due to mineralization of both RPEC and soil organic P, and chelation of P through ligand exchange reactions to reduce P fixation throughout the crop growth stages. The ligand exchange reactions can increase P mobilization through organic and phosphate anions adsorption with Fe and Al sites [72]. Slow release of P through mineralization of organic P fraction from enriched compost was reported previously [73]. Organic acids produced by phosphate solubilizing microorganisms are sources of $H^+$ ions which help mineralize tri-calcium phosphate of RP to mono-calcium phosphate; the available form of phosphorus for better plant growth [74].

The application of compost treatments showed significantly higher nitrate-nitrogen contents in post-harvest soil compared to control. Higher nitrate nitrogen content from compost treated plots would be due to reduced nitrate leaching from the soil [75]. Sommers and Giordano [76] stated that all inorganic N in soil amended with Municipal Solid Waste (MSW) compost was available for plant uptake, but 5 to 75% of the organic N will be mineralized within 1 year after application. The findings are in accordance with the results of Baziramakenga et al. [77] who reported an increase in inorganic nitrogen ($NO_3$-N) contents of snap-bean post-harvest soil with the application of compost of de-inking paper residues and poultry manure. The reason for higher $NO_3$-N contents due to the application of composts is attributed to the formation of phospho-protein due to the interaction with rock phosphate from enriched compost, which is less susceptible to volatilization. The proteins are decomposed by soil bacteria and change into ammonium that is further nitrified by nitrifying bacteria. This form of nitrogen from compost is slowly available to plants having fewer chances of loss through volatilization. The escape of ammonia from soil decreases if the nitrogen source is compost, organic manure or green manure [78]. The presence of phosphate preserves the nitrogen resulting in a decrease in the number of denitrifying bacteria [79]. The slow release process of nutrients from enriched compost might be another reason for higher nitrate-nitrogen contents than inorganic fertilizers (FDP) in post-harvest soil. Adeli et al. [80] reported higher residual soil $NO_3$-N contents after cotton crops with the application of poultry manure compared to inorganic fertilizers. Seed inoculation with PGPR (*Pseudomonas* sp. and *Proteus* sp.) showed an increase in post-harvest nitrate-nitrogen contents. Canbolat et al. [81] also reported increase in soil post-harvest nitrogen contents with application of *Pseudomonas putida* compared to the control on barley crop.

Extractable potassium contents increased in post-harvest soil with the application of compost compared to inorganic P fertilizer (FDP). A significantly higher concentration of K with the application of enriched compost compared to FDP and control might be due to the higher water-soluble potassium present in the enriched composts. Stratoon et al. [82] reported that K in composts remains in water soluble forms and thus does not need to be mineralized before becoming available to plants. The increase in soil extractable K by rock phosphate enriched compost (RPEC) may be related to the direct addition to the available K pool of the soils, and to the reduction of K fixation and increase the release of K from the soil solid phase due to the interaction of organic matter and/or soil microorganisms with K-bearing minerals [77]. It was revealed that potassium in manure and compost is highly plant-available and can be used similar to K fertilizer application [83].

Soil enzymes (alkaline phosphatase and acid phosphatase) play a vital role in conversion of fixed soil phosphorus to plant available form [7]. The increase in alkaline phosphatase activities with the application of RP enriched compost in the present study may be due to the availability of organic C which consequently increased the soil phosphatase activity [84] and the compost might have provided considerable carbon and nitrogen for maximum growth of microbes. It has been emphasized that C and N are interlinked with P mineralization by microbes [85] and Shrivastava et al. [10]

concluded that the availability of metabolizable C plays a significant role to increase soil phosphatase activity with the application of P enriched manure on mungbean crop. Soil enzymes such as acid and alkaline phosphatases help to increase mineralization of $P_0$ to $P_i$ by creating a strong relation between bio-available and unavailable P in the soil [86]. Some researchers [87,88] believe that there is an inverse relationship between available P and phosphatases due to negative feedback of phosphate ions on PHO genes, that suppress phosphatase synthesis by microbes [89]. However, phosphatase activity was not affected by the use of rock phosphate as a phosphate source in RP enriched compost [9] showing long persistence and least biodegradation of enzymes with the application of compost. But Pascual et al. [90] endorsed the decrease in phosphatase activities with time span due to exhaustion of biodegradable substrates by microbial activity.

Microbial biomass is an important factor assessing soil quality and its ability to provide energy for nutrient recycling and transformation in the soils [91]. Kiani et al. [92] documented the microbial biomass responses to different land management systems including fertilizer addition and organic amendment application and identified suitable soil quality indicators. The microbial biomass carbon (MBC) acts as substrate supplying entity for microbial communities in soil [93]. In the present study, maximum MBC with the application of RP enriched compost compared to poultry litter compost is due to higher percentage of existing microbial biomass carbon, mineralizable nitrogen and water-soluble carbon in the former compost. The results are in conformity to the findings of Meena et al. [94] who reported an increase in soil microbial biomass carbon with the application of enriched compost compared to ordinary compost as well as inorganic fertilizer. Previously Ayed et al. [95] found an increase in microbial biomass carbon with the application of compost compared to inorganic chemical fertilizer and control in wheat crop.

The maximum microbial biomass phosphorus (MBP) with the application of RPEC and inorganic P fertilizer could be due to the transformation of labile and nonlabile inorganic phosphorus to the organic pool through microbial activity of compost. As Leytem et al. [96] reported the assimilation of various fractions of P into microbial biomass which ultimately provides available P for plants since most organic phosphorus in microbial cells is hydrolysable. Microbial biomass phosphorus may also help in calcareous soils by providing plant available P with application of manure [97] by the mechanism in which P is immobilized and transformed to labile P, which is safe from fixation and transmitted to available P [98]. The results in the present study showed less microbial biomass P with the addition of a full recommended dose of inorganic P fertilizer, indicating that soil existing organic carbon was limited to support microbial growth and activity. Minimum microbial biomass was recorded with the addition of high P inorganic fertilizer to the soil in an incubation study.

## 5. Conclusions

The present research revealed that enrichment of rock phosphate with poultry litter and PGPR during the process of composting improves nutrient availability and biological properties of the compost. Application of RP enriched compost in field experiment increased yield and yield components of the wheat crop compared to the full recommended dose of inorganic fertilizer and control. Moreover, seed inoculation with PGPR showed significant results to improve the agronomic effectiveness of RP enriched compost. Chemical (availability of phosphorus) and biological (microbial biomass C & P, alkaline and acid phosphatase activities) properties of post-harvest soil improved with the application of RP enriched compost. It can be concluded that RP enriched compost may be an alternative to chemical fertilizer to improve the growth and yield of the crop.

**Author Contributions:** Conceptualization, M.B., A.B. and N.K.; Methodology, M.B., M.K., K.M.D., and N.K.; Formal analysis, M.B., M.K., S.N. and N.K.; Software, M.B., K.M.D. and N.K.; Data curation, S.N., K.M.D. and A.M.; Validation, M.B., A.B., A.H., A.M. and N.K., Investigation, M.B., A.B., S.B., A.H. and N.K., Resources, M.B., M.K., A.B., S.N., A.M. and N.K., writing—original draft preparation, M.B., M.K., A.B. and N.K.; writing—review and editing, M.B., M.K., A.B., A.H., A.S. and N.K.; supervision, M.B., A.B. and N.K. All authors have read and agreed to the published version of the manuscript.

**Funding:** This research received no external funding.

**Acknowledgments:** The author is thankful to Higher Education Commission Islamabad Pakistan and Pakistan Agriculture Research Council, Islamabad, Pakistan for the assistance to conduct this research.

**Conflicts of Interest:** The authors declare no conflict of interest.

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
