# Peer review of "Rock Phosphate-Enriched Compost in Combination with Rhizobacteria; A Cost-Effective Source for Better Soil Health and Wheat (Triticum aestivum) Productivity"

_agronomy, doi:10.3390/agronomy10091390_

Round 1

Reviewer 1 Report

Dear Authors:

Thank you for your submission examining integrated nutrient management effects on soil health and wheat productivity. This manuscript reports the results of a 2-year experiment on using various fertilizer sources with and without inoculation with beneficial bacteria on plant and soil parameters. Overall, the experiment was well-designed and the results appear to support the conclusions. I recommend resubmission after major revision with high probability of acceptance pending response to the following comments and suggestions. Please consider the following:

1) Improvement of the introduction to better connect the treatments with the phytohormone effects. Lines 407-425 discuss these effects, but it would be interesting to see them presented in the introduction and how they link to productivity from the different treatments.

2) The data presented are reported as means of the two years. This is suitable if there were not interactions between the two years. The model used should include year as a random effect. When year does not interact with either of the main or split treatments than the way the results are presented is valid. There is no mention of accounting for the year in the statistical model.

3) Data presented in Figures 1, 2 and 3 do not show p values with statistical statements nor Tukey mean separations. Based on the statistical model presented it appears the same approach was used for the tables and the figures. It's unclear why the data are presented differently.

4) The discussion of nitrate-N is incomplete. Higher residual nitrate suggests the soil samples were collected after the wheat crop was harvested. Note there is no mention of when these soil samples were collected. Line 128 mentions initial soil samples, but residual nitrate is a term used for soil measurements post harvest and/or between crop rotations. Nevertheless, higher residual nitrate suggests the N was used inefficiently. Given the focus of the paper does not include a comprehensive assessment of N cycling, the authors must acknowledge higher residual nitrate may be the result of mis-timed fertilizer application and N mineralization from organic-based N sources in a manner lead to more N available then needed. In fact, lower residual nitrate would indicate N was more efficiently used by the wheat crop.

Pleas consider these major issues and resubmit.

Sincerely,

Reviewer

Author Response

Thank you for your submission examining integrated nutrient management effects on soil health and wheat productivity. This manuscript reports the results of a 2-year experiment on using various fertilizer sources with and without inoculation with beneficial bacteria on plant and soil parameters. Overall, the experiment was well-designed and the results appear to support the conclusions. I recommend resubmission after major revision with high probability of acceptance pending response to the following comments and suggestions. Please consider the following:

Response: We would like to thank the Reviewer for his/her evaluation and for the constructive comments and suggestions that have helped us improve the quality of the manuscript. We have revised the manuscript following your suggestions and comments to improve its quality. All the changes made in responses to the Reviewer’s comments are tracked in the revised MS file. We hope that our revised version will now meet your expectations. Please see below our responses itemized to your comments and suggestions.

1) Improvement of the introduction to better connect the treatments with the phytohormone effects. Lines 407-425 discuss these effects, but it would be interesting to see them presented in the introduction and how they link to productivity from the different treatments.

Response: We are very thankful to the reviewer for this useful comment. Based on this comment we have modified the Introduction and the link has been developed by incorporating the sentences given at lines 80-92, page number 2&3.

2) The data presented are reported as means of the two years. This is suitable if there were not interactions between the two years. The model used should include year as a random effect. When year does not interact with either of the main or split treatments than the way the results are presented is valid. There is no mention of accounting for the year in the statistical model.

Response: We highly appreciate the Reviewer for this constructive comment and we agree with the reviewer. All these required sentences have been incorporated in the MS (lines 167-168 page number 5).

3) Data presented in Figures 1, 2 and 3 do not show p values with statistical statements nor Tukey mean separations. Based on the statistical model presented it appears the same approach was used for the tables and the figures. It's unclear why the data are presented differently.

Response: The required data has been added and Tukey HSD values have been incorporated at the figures legend.

4) The discussion of nitrate-N is incomplete. Higher residual nitrate suggests the soil samples were collected after the wheat crop was harvested. Note there is no mention of when these soil samples were collected. Line 128 mentions initial soil samples, but residual nitrate is a term used for soil measurements post harvest and/or between crop rotations. Nevertheless, higher residual nitrate suggests the N was used inefficiently. Given the focus of the paper does not include a comprehensive assessment of N cycling, the authors must acknowledge higher residual nitrate may be the result of mis-timed fertilizer application and N mineralization from organic-based N sources in a manner lead to more N available then needed. In fact, lower residual nitrate would indicate N was more efficiently used by the wheat crop.

Response: We are thankful to the reviewer for these important suggestions. The missing discussion part has been incorporated in the MS (lines 494-497, page number 15)

Reviewer 2 Report

The manuscript assessed the effect of enriched poultry manure litter combined with PGPRs on P availability from RP and its comparative effectiveness with mineral fertilizer on soil nutrient status, wheat growth and production. Authors present data from grain yield, dry matter yield, leaf chlorophyll and phytohormones (IIA and GA) concentration and soil nutrient status (available P, nitrate N, extractable K, microbial biomass C and P, and alkaline phosphatase). The conclusion application of RP enriched compost improved wheat yield and biochemical soil properties in post-harvest. In general, the manuscript was well written and clear. The manuscript is interesting due to the future phosphorus reserves depletion threat food security. Accordingly, studies related to find alternatives, such as mixing RP with compost and PGPRs, to decrease inorganic P fertilizers demand are increasing.

In the general objective was mentioned a comparison between the enriched compost with inorganic fertilizers, although only SSP (full and half dose) was used. In addition, evaluation of P availability from RP enriched compost, but the study only present data of P availability from the soil amended and plant P content (P uptake and Seed P content). Please clarify that in the objective.

Treatments were difficult to follow, in the beginning was mentioned that Control is untreated and un-inoculated, SPLC is poultry litter compost, RP rock phosphate, RPEC1 is RP combined with poultry litter solubilized with Pseudomonas sp.; RPEC2 RP poultry litter with Proteus sp.; FDP as full SSP and HDP as of half dose of SSP. However, in the results is presented each treatment with inoculation of Pseudomonas and Proteus and un-inoculated.

Line 63. Add the reference please

Line 85. What kind of soil was used on this study? (Soil type)

Line, 94. N was applied as urea or compost on nutrient basis. Did you calculate how much of poultry litter was added to supply the 100 g N ha-1?

Line 126. Please specify how did you calculated plant P uptake. Only shoot, root, or both were considered to calculate it?

Line 191. Figure 1. When you use ug g-1 is concentration not content. Please correct using concentration instead of content

Line 204. Figure 2. Please correct IIA concentration instead of chlorophyll content

Line 217. Figure 3. Please correct using concentration instead of content

In the Figures, could you add the letter of significance?

Author Response

The manuscript assessed the effect of enriched poultry manure litter combined with PGPRs on P availability from RP and its comparative effectiveness with mineral fertilizer on soil nutrient status, wheat growth and production. Authors present data from grain yield, dry matter yield, leaf chlorophyll and phytohormones (IIA and GA) concentration and soil nutrient status (available P, nitrate N, extractable K, microbial biomass C and P, and alkaline phosphatase). The conclusion application of RP enriched compost improved wheat yield and biochemical soil properties in post-harvest. In general, the manuscript was well written and clear. The manuscript is interesting due to the future phosphorus reserves depletion threat food security. Accordingly, studies related to find alternatives, such as mixing RP with compost and PGPRs, to decrease inorganic P fertilizers demand are increasing.

Response: Response: We would like to thank the Reviewer for his/her evaluation and for the constructive comments and suggestions that have helped us improve the quality of the manuscript. We have revised the manuscript following your suggestions and comments to improve its quality. All the changes made in responses to the Reviewer’s comments are tracked in the revised MS file. We hope that our revised version will now meet your expectations. Please see below our responses itemized to your comments and suggestions.

In the general objective was mentioned a comparison between the enriched compost with inorganic fertilizers, although only SSP (full and half dose) was used. In addition, evaluation of P availability from RP enriched compost, but the study only present data of P availability from the soil amended and plant P content (P uptake and Seed P content). Please clarify that in the objective.

Response:  We are thankful to the reviewer for this important suggestion.The amendments are done in the objective at line 98, page number 3.

Treatments were difficult to follow, in the beginning was mentioned that Control is untreated and un-inoculated, SPLC is poultry litter compost, RP rock phosphate, RPEC1 is RP combined with poultry litter solubilized with Pseudomonas sp.; RPEC2 RP poultry litter with Proteus sp.; FDP as full SSP and HDP as of half dose of SSP. However, in the results is presented each treatment with inoculation of Pseudomonas and Proteus and un-inoculated.

Response: We are thankful to the reviewer for this important comment. The amendments are done in the results section and the word “control” has been corrected according to the already mentioned term “ un-inoculated untreated control” as in the start of the manuscript, the changes are done at line 174, 175, 191, 192, 199,200, 224, 236, 252, 264, 265, 280, 302,303, 312, 320,321, 349, 352, 353.

Line 63. Add the reference please

Response: A reference has been added at the end of these lines.

Line 85. What kind of soil was used on this study? (Soil type)

Response: Silty loam soil was used in the experiments. These information are also added in the MS at lines 104-105, page number 3

Line, 94. N was applied as urea or compost on nutrient basis. Did you calculate how much of poultry litter was added to supply the 100 g N ha-1?

Response: We are thankful to the reviewer for these important comments. All these required information are added to the MS at lines 116-117.

Line 126. Please specify how did you calculated plant P uptake. Only shoot, root, or both were considered to calculate it?

Response: Based on these comments, we have added an explanation at lines 147-48.

Line 191. Figure 1. When you use ug g-1 is concentration not content. Please correct using concentration instead of content

Response: This has been corrected as suggested by the reviewer.

Line 204. Figure 2. Please correct IIA concentration instead of chlorophyll content

Response: This has been corrected as suggested by the reviewer.

Line 217. Figure 3. Please correct using concentration instead of content

Response: This has been corrected as suggested by the reviewer.

In the Figures, could you add the letter of significance?

Response: We have added the letters of significance to all figures.

Reviewer 3 Report

1) The Abstract  exceed  200 words ( 266 word in fact). The authors must respect the Journal Requests. 

2) All Chapters must have different numbers. Actually, in tet all Chapter and Sub chapter ( subsection) have the same number (i.e. 1 or 1.1 or 1.1.1); these must be different (see the papers previously published in Agronomy - Basel as a model). Example:

1.Introduction

2. Material and methods

3. Results

4. Discussion

5. Conclusions

3) In table 2, the unit of measure for Tillers is adecvately?  In the  table 2 appear: ''   Tillers (m-2)''. What represent the Tillers in this table? A volume, a surface or  a number ?  The measure unit must considered in connection with this explanation.

4)In table 2 and in table 5, at all values , appears at superscript a lot of letters. Perhaps this represent the results returned from statistical analysis.  If this supposition are correct, then authors must  provide explanations at Methodology. In the case in which one ore more of  letters from superscripts represent experimental variants with statistical significance, this fact must be mentioned at Methodology.

4) At Methodology: All Sub chapters (i.e. subsections) has  the same number (i.e. 1.1). If the authors choose to put numbers at subsections, then these numbers must be different,  in mathematical order, i.e.:

2.1  Experimental site and treatments

2.2. Seed inoculation

2.3. Yield, physiology and plant nutrient analysis

2.4. Soil analysis

2.5. Statistical analysis

-The authors must to specify the surface (plot, in m2 or in ha) used in their experiments, for each experimental variants.

-The authors must clearly specify at methodology if in all experimental variants (21 experimental variants), the  fertilizations were made in the same time with sowing;
-The authors must to specify the norm of sowing used in all experiments.

4) At the chapters  ''Results'' an at the Chapter ''Discussion'': the same observations. Example: 

3. Results

3.1. Yield  and yield components

3.2. Leaf chlorophyll, IAA and GA contents

3.3. Plant phosphorus uptake and seed phosphorus

3.4. Soil Properties

3.4.1.Available P, nitrate nitrogen and extractable potassium

3.4.2. Alkaline phosphatase and microbial biomass

3.4.3.Economic analysis

Author Response

We would like to thank the Reviewer for his/her evaluation and for the constructive comments and suggestions that have helped us improve the quality of the manuscript. We have revised the manuscript following your suggestions and comments to improve its quality. All the changes made in responses to the Reviewer’s comments are tracked in the revised MS file. We hope that our revised version will now meet your expectations. Please see below our responses itemized to your comments and suggestions.

The Abstract  exceed  200 words ( 266 word in fact). The authors must respect the Journal Requests.

Response: The abstract length has been decreased to 194 words.  

2) All Chapters must have different numbers. Actually, in tet all Chapter and Sub chapter ( subsection) have the same number (i.e. 1 or 1.1 or 1.1.1); these must be different (see the papers previously published in Agronomy - Basel as a model). Example:

1.Introduction

  1. Material and methods
  2. Results
  3. Discussion
  4. Conclusions

Response: We have corrected the subheading numbers.

3) In table 2, the unit of measure for Tillers is adecvately?  In the  table 2 appear: ''   Tillers (m-2)''. What represent the Tillers in this table? A volume, a surface or  a number ?  The measure unit must considered in connection with this explanation.

Response: The correction is done in heading of table 4 as “Number of tillers m-2

4) In table 2 and in table 5, at all values , appears at superscript a lot of letters. Perhaps this represent the results returned from statistical analysis.  If this supposition are correct, then authors must  provide explanations at Methodology. In the case in which one ore more of  letters from superscripts represent experimental variants with statistical significance, this fact must be mentioned at Methodology.

Response: We are thankful to the reviewer for this important suggestion. We have already added an explanation to all these different letters in the table legends.

4) At Methodology: All Sub chapters (i.e. subsections) has  the same number (i.e. 1.1). If the authors choose to put numbers at subsections, then these numbers must be different,  in mathematical order, i.e.:

2.1  Experimental site and treatments

2.2. Seed inoculation

2.3. Yield, physiology and plant nutrient analysis

2.4. Soil analysis

2.5. Statistical analysis

Response: The subheading numbers have been revised as suggested.

-The authors must to specify the surface (plot, in m2 or in ha) used in their experiments, for each experimental variants.

Response: We are thankful to the reviewer for this important comment. The suggested amendment is incorporated to the MS at line 115.

-The authors must clearly specify at methodology if in all experimental variants (21 experimental variants), the  fertilizations were made in the same time with sowing;
-The authors must to specify the norm of sowing used in all experiments.

Response: Based on these comments, We have added an explanation in the MS at lines 117-118.

4) At the chapters  ''Results'' an at the Chapter ''Discussion'': the same observations. Example: 

  1. Results

3.1. Yield  and yield components

3.2. Leaf chlorophyll, IAA and GA contents

3.3. Plant phosphorus uptake and seed phosphorus

3.4. Soil Properties

3.4.1.Available P, nitrate nitrogen and extractable potassium

3.4.2. Alkaline phosphatase and microbial biomass

3.4.3.Economic analysis

Response: We followed the same pattern in different chapters.

Reviewer 4 Report

Review for

Rock Phosphate-Enriched Compost in Combination with PGPR; a Cost-Effective Source for Better Soil Health and Wheat (Triticum aestivum) Productivity

agronomy-888624

This study showcases linkages of phosphorus supply and bio-stimulants in plant nutrition.

To include the abbreviation of “plant growth promoting rhizobacteria” in the title reduces the understandability of the title. Please considering revising the title.

L21 “The present study”

L43 “destruction of” sounds overshot. Please try “suppression of “ or “hindering”

L59 The sentence “Isolated phosphate-solubilizing … … soil P availability under pot conditions” is long and confusing. Please revise.

The title highlights “Soil Health”; however, this aspect needs more development in the paper. Suggest to look into Kiani et al. (2017). Quantifying sensitive soil quality indicators across contrasting long-term land management systems: crop rotations and nutrient regimes. Agric. Ecosyst. Environ. 248:123–135. Kiani et al. (2017) documented microbial biomass responses to management including fertilizer addition and organic amendment application (related to L 509). Other soil quality indicators were assessed as related to nutrient cycling and availability for plants, including also water availability (related to L 377) and soil structure (linked to statement in L44). This will provide more context to the intro and discussion.

Table 3. there is a need to add the units of EC. By the way, is this electric conductivity? Please clarify.

Table 4. L143 states the usage of ANOVA followed by Tukey; however, Table 4 shows LSD. This inconsistency needs to be clear out. What are the values within parentheses in Table 4? These values are extremely low to show variability of three field replicates. This needs to be clarified – how were these values calculated? For example, 4 to 16 kg grainDM ha-1 of variability is unheard of.

L272 consider “collected two days after wheat harvesting”

Table 6. It will be good to specify somewhere in the text that Rs. refer to the national currency (rupees).

Author Response

Rock Phosphate-Enriched Compost in Combination with PGPR; a Cost-Effective Source for Better Soil Health and Wheat (Triticum aestivum) Productivity

This study showcases linkages of phosphorus supply and bio-stimulants in plant nutrition.

Response: We would like to thank the Reviewer for his/her evaluation and for the constructive comments and suggestions that have helped us improve the quality of the manuscript. We have revised the manuscript following your suggestions and comments to improve its quality. All the changes made in responses to the Reviewer’s comments are tracked in the revised MS file. We hope that our revised version will now meet your expectations. Please see below our responses itemized to your comments and suggestions.

To include the abbreviation of “plant growth promoting rhizobacteria” in the title reduces the understandability of the title. Please considering revising the title.

Response: We have removed the acronym from the title.

L21 “The present study”

Response: This line has been corrected.

L43 “destruction of” sounds overshot. Please try “suppression of “ or “hindering”

Response: the word destruction has been replaced with suppression.

L59 The sentence “Isolated phosphate-solubilizing … … soil P availability under pot conditions” is long and confusing. Please revise.

Response: This line has been rephrased as ‘’ Isolated phosphate-solubilizing fungi from phosphate mines of China were reported to have efficient biofertilizers and P solubilizers with the capacity to enhance growth of wheat.’’

 The title highlights “Soil Health”; however, this aspect needs more development in the paper. Suggest to look into Kiani et al. (2017). Quantifying sensitive soil quality indicators across contrasting long-term land management systems: crop rotations and nutrient regimes. Agric. Ecosyst. Environ. 248:123–135. Kiani et al. (2017) documented microbial biomass responses to management including fertilizer addition and organic amendment application (related to L 509). Other soil quality indicators were assessed as related to nutrient cycling and availability for plants, including also water availability (related to L 377) and soil structure (linked to statement in L44). This will provide more context to the intro and discussion.

Response: The highlighted sections have been modified. We added the proposed study to our MS as ‘’ Kiani et al. [93] documented the microbial biomass responses to different land management systems including fertilizer addition and organic amendment application and identified suitable soil quality indicators (L 511-13).

Table 3. there is a need to add the units of EC. By the way, is this electric conductivity? Please clarify.

Response: Yes it stands for electrical conductivity. The unit for Ec was added to the MS as (dSm-1).

Table 4. L143 states the usage of ANOVA followed by Tukey; however, Table 4 shows LSD. This inconsistency needs to be clear out. What are the values within parentheses in Table 4? These values are extremely low to show variability of three field replicates. This needs to be clarified – how were these values calculated? For example, 4 to 16 kg grainDM ha-1 of variability is unheard of.

Response: The suggested amendments are done and LSD has been replaced with HSD in each table and the standard errors in parenthesis have been corrected in table 4.

L272 consider “collected two days after wheat harvesting”

Response: The line has been rephrased.

Table 6. It will be good to specify somewhere in the text that Rs. refer to the national currency (rupees).

Response: A note has been added at the table legend.

Round 2

Reviewer 1 Report

Congratulations on these edits. Paper is ready for publication.

Author Response

We are thankful to the reviewer for recommending our paper for publication.

Reviewer 4 Report

The paper has been improved based on the provided input.

Author Response

We are thankful to the reviewer for his/her positive feedback on our MS.